# Management of Peritoneal Metastasis in Patients with Pancreatic Ductal Adenocarcinoma

**DOI:** 10.3390/curroncol32020103

**Published:** 2025-02-12

**Authors:** Grace Wu, Oliver J. Standring, Daniel A. King, Sepideh Gholami, Craig E. Devoe, Cornelius A. Thiels, Travis E. Grotz, Matthew J. Weiss, Richard L. Whelan, Mustafa Raoof, Danielle K. DePeralta

**Affiliations:** 1Northwell Health, New Hyde Park, NY 11040, USA; gwu3@northwell.edu (G.W.); ostandring@northwell.edu (O.J.S.); dking14@northwell.edu (D.A.K.); sgholami@northwell.edu (S.G.); cdevoe@northwell.edu (C.E.D.); mweiss8@northwell.edu (M.J.W.); rwhelan1@northwell.edu (R.L.W.); 2Department of Surgery, Zucker School of Medicine at Hofstra/Northwell, Manhasset, NY 11030, USA; 3Northwell Health Cancer Institute, Lake Success, NY 11042, USA; 4Department of Surgery, Mayo Clinic, Rochester, MN 55905, USAgrotz.travis@mayo.edu (T.E.G.); 5Department of Surgery, City of Hope National Medical Center, Duarte, CA 91010, USA; mraoof@coh.org

**Keywords:** pancreatic ductal adenocarcinoma (PDAC), peritoneal metastases, intraperitoneal chemotherapy, cytoreductive surgery (CRS), heated intraperitoneal chemotherapy (HIPEC), pressurized intraperitoneal aerosolized chemotherapy (PIPAC)

## Abstract

The peritoneum is the second most common site of metastasis in patients with pancreatic ductal adenocarcinoma (PDAC). Up to half of all patients that undergo curative-intent resection eventually develop peritoneal metastasis (PM), which accounts for significant morbidity and drives mortality. Despite recent advances in management, PM is associated with very poor prognosis, which is often measured in weeks to months. Clinical manifestations including bowel obstruction, ascites, and urinary obstruction have profound impact on quality of life. Even with relatively advanced disease, PM often remains occult on imaging and thus tend to be underdiagnosed and understudied. Many patients with peritoneal-only PM are excluded from clinical trials because response cannot be measured by standard radiographic criteria. Furthermore, as patients with PM are not eligible for surgical resection and low-volume peritoneal disease is often not amenable to percutaneous biopsy, tissue samples for peritoneal-specific translational studies are limited. Intraperitoneal therapeutics have been proposed as an attractive option for PM, as better penetration of tumor tissue can be achieved with less systemic toxicity compared with intravenous chemotherapy. Heated intraperitoneal chemotherapy (HIPEC), typically combined with cytoreductive surgery (CRS), is an option for select patients with PM from gynecologic or gastrointestinal primary, and for patients with primary peritoneal mesothelioma. However, the incorporation of locoregional therapy for PM in patients with PDAC has been poorly studied given the aggressive nature of pancreatic cancer and overall poor prognosis. With recent advances in existing treatment options, there may be a subset of patients who may derive benefits from locoregional control with cytoreduction and/or intraperitoneal chemotherapy. Critically, additional work is needed to determine PM-favorable clinical and tumoral predictive biomarkers to identify patients who may benefit from a more aggressive approach. We describe the current state of management of patients with peritoneal metastasis from PDAC and review the available data exploring peritoneal-directed therapy with cytoreductive surgery and/or intraperitoneal chemotherapy.

## 1. Introduction

Pancreatic ductal adenocarcinoma (PDAC) remains highly lethal, with a 5-year survival rate of 8% despite recent advances in treatment [1]. Approximately half of all patients present with metastatic disease, with the peritoneum representing the second most common site of metastasis after the liver [2]. While only 10% of patients present with synchronous peritoneal metastasis (PM), up to 50% of patients develop metachronous PM after curative-intent resection, and autopsy studies report PM in 50% of patients at the time of death [3]. Additionally, 20–30% of PDAC patients present with positive cytology in the absence of overt PM [3], highlighting a high rate of radiographic-occult disease.

The burden of peritoneal disease frequently drives poor quality of life and mortality associated with PDAC, with high rates of bowel/urinary obstruction, ascites, pain, and acceleration of cancer-associated weight loss [4,5]. Recurrent malignant bowel obstruction is a common event, with an incidence as high as 28% [6]. Limited treatment options often require hospitalization, where time spent in the hospital relative to remaining life is considerable [6]. This dramatically impacts quality of life in the weeks to months prior to death, while being an important cause of mortality [4,7].

Despite the prevalence of PM and the associated morbidity, little is known about the tumor biology of patients with PM from PDAC, especially those with peritoneal-only metastasis. Biologically, PMs demonstrate a distinct molecular signature compared to hematogenous or lymphatic spread, associated with increased ability to shed from the primary tumor and adhere to the surrounding peritoneum [8]. Direct spread from intraperitoneal organs may therefore explain the burden of disease from pancreatic tail lesions [4]. However, the impact of differential tumor biology and treatment response is not well known. Response to therapy may or may not mimic the primary tumor, adding an additional therapeutic challenge. Peritoneal-only metastasis represents a “regional” subtype of stage IV disease and in select patients there may be an opportunity for a more aggressive locoregional approach to therapy [9].

Preclinical research suggests that specific genes govern the discrete steps of the metastatic cascade of PM, including those regulating cell adhesion, epithelial to mesenchymal transition (EMT), apoptosis, and tumor invasion, which are distinct from those involved in hematogenous metastases [10]. These mediators allow cancer cells to detach from the primary tumor and survive in the peritoneal fluid, where they are transported to sites along the peritoneal surface. They must then attach to the peritoneum, with subsequent invasion into underlying tissues to create viable metastatic deposits (Figure 1). Each step in the metastatic cascade presents an opportunity for therapeutic intervention.

Drug delivery to peritoneal metastases is a major challenge. In 1996, Jacquet and Sugarbaker described the “peritoneal–plasma barrier” which limits penetration of many chemotherapeutics with intravenous or oral routes of administration [11]. Normothermic intraperitoneal chemotherapy (NIPEC) has been explored in pancreatic cancer with some efficacy signal [12]. However, heated intraperitoneal chemotherapy (HIPEC) is traditionally combined with cytoreductive surgery (CRS) and is considered the standard of care in select gastrointestinal or gynecological malignancies [13]. A small number of early studies evaluating the safety and feasibility of cytoreductive surgery with HIPEC in PDAC reported encouraging results [14]. In addition, an emerging alternative to HIPEC is pressurized intraperitoneal aerosolized chemotherapy (PIPAC), which uses a nebulizer to convert liquid chemotherapy into an aerosol. Some efficacy in pancreatic cancer PM has been suggested, with initial evidence of carcinoma regression in 50–80% of patients [15]. Intraperitoneal therapies under active research are summarized in Figure 2.

Peritoneal metastases are challenging to detect on cross-sectional imaging. Despite improvements in axial imaging with high quality CT and MRI, detection of PM in traditional imaging modalities is still difficult due to their small size and limited resolution of soft tissue [16]. Compared to the peritoneal carcinomatosis index (PCI; Figure 3), determined by direct visualization, CT may undervalue the presence of peritoneal metastases by as much as 20–30% [17]. This leads to underreporting of PM in national databases and makes it difficult to assess treatment response without repeated laparoscopy, which is not feasible in most cases [18]. Despite their prevalence, patients with peritoneal-only metastasis are frequently excluded from clinical trials because they do not have measurable disease by RECIST criteria [17]. Most patients are never considered for more aggressive locoregional approaches even in the setting of limited peritoneal metastasis. The presence of positive intraperitoneal cytology in otherwise resectable patients represents an area of therapeutic uncertainty and is associated with poor prognosis [19]. However, recent studies suggest those who undergo neoadjuvant therapy and convert to negative cytology prior to curative-intent surgery have significantly improved survival [20]. Whether intraperitoneal chemotherapy could be incorporated into a localized setting to increase peritoneal control and improve survival remains an enticing area of study.

## 2. Peritoneal-Directed Therapy with Cytoreductive Surgery, HIPEC, PIPAC, and NIPEC

### 2.1. Curative-Intent Heated Intraperitoneal Chemothreapy (HIPEC): Adjuvant Therapy in the Setting of Resected Localized Disease

HIPEC is frequently combined with cytoreductive surgery (CRS) and although controversial, is considered by many to be the standard of care in several gastrointestinal or gynecological malignancies. This modality combines heat with liquid intraperitoneal chemotherapy.

In PDAC, the published literature includes four studies that selected patients with localized PDAC and no evidence of PM in the adjuvant setting with the goal of mitigating the risk of peritoneal relapse (Table 1). This includes single-arm studies from Sugarbaker et al., Yurttas et al., Tentes et al., and a randomized phase II/III study by Padilla-Valverde et al. for a total of 88 patients [21,22,23,24]. In all three studies, only patients with resectable PDAC without evidence of distant metastases, and no prior systemic chemotherapy were included. Patients with unresectable primary tumors or occult PM were excluded from analysis. Immediately following curative-intent primary resection, all patients underwent HIPEC perfusion with 1000 mg/m^2^ gemcitabine for 60 min in the three studies with single-arm analyses. In the phase II–III trial, patients in the control arm (no HIPEC) received individualized adjuvant treatment, and those in the experimental arm received HIPEC with 120 mg/m^2^ gemcitabine for 30 min. In all cases, HIPEC was performed prior to reconstruction of gastrointestinal continuity to ensure that the resection bed was fully exposed to chemotherapy.

The first single-arm study examined the safety of intraperitoneal gemcitabine in the context of both HIPEC and subsequent NIPEC [21]. After undergoing pancreatoduodenectomy and HIPEC, an indwelling port was placed to allow for repeated intraperitoneal chemotherapy. NIPEC was then performed about 6 weeks postoperatively once a week with 1000 mg/m^2^ gemcitabine for six cycles. The study enrolled 12 patients who underwent pancreaticoduodenectomy and HIPEC, and 8 of 12 were eligible for NIPEC. One of the eight declined further treatments, with seven patients completing the protocol. For the eight patients who were treated with HIPEC + NIPEC (including the one who did not complete all NIPEC sessions), the median overall survival was 29 months from resection + HIPEC. Notably, no patients that completed the protocol developed local recurrence or peritoneal metastases, suggesting that intraperitoneal chemotherapy may afford some degree of locoregional control despite distant failure. With no increase in operative morbidity or mortality (four minor morbidities, one grade III pancreatic fistula requiring percutaneous drainage, no returns to the operating room), the authors concluded gemcitabine is safe to use as an intraperitoneal chemotherapy agent when combined with the Whipple procedure.

Similarly, Yurttas et al. aimed to evaluate the safety of pancreatectomy with gemcitabine monotherapy administered as HIPEC in 16 patients [22]. No short-term mortality occurred after 30 and 90 days. Using the National Cancer Institute Common Terminology Criteria for Adverse Events (CTCAE) version 4.0, there were no adverse events greater than grade III (severe), with the majority of adverse events occurring as grade I (mild) or grade II (moderate), specifically nausea, vomiting, or diarrhea. In terms of surgical complications, 3 among 13 patients under risk developed pancreatic fistulas (i.e., excluding those who underwent total pancreatectomy), comparable to rates reported in the literature. While the study was only designed for safety and feasibility, supplementary oncologic data was collected with a median OS of 16.1 months after surgery and a 1-year survival rate of 62.5%. Overall, the safety and feasibility of pancreatectomy combined with HIPEC was deemed to be comparable to resection alone.

In 2021, Tentes et al. updated findings from a study initially published in 2016 [23,25], in which 39 patients underwent R0 resection with concomitant HIPEC-gemcitabine from 2007 to 2018. Hospital mortality occurred for two patients (5.1%), secondary to pulmonary failure and sepsis. Surgical complications included one case of postoperative bleeding requiring re-operation and four cases of choledochojejunal anastomosis failure. Two required re-operation with T-tube insertion; the other two were treated conservatively. Twenty-three patients (59%) developed recurrence, with 19 (48.7%) cases of distant metastatic disease and 4 (10.3%) locoregional. No patients who developed locoregional recurrence also developed distant metastatic disease. Five-year overall survival was 24%, with the 1, 2, and 3-year survival rates at 64%, 38%, and 28%, respectively. The median OS was 17 months and the median disease-free survival was 11 months. These results suggest HIPEC may be useful in reducing locoregional failure, but overall survival remains poor.

Padilla-Valverde et al. reported the first randomized clinical trial comparing surgical resection with curative intent followed by investigator-choice adjuvant chemotherapy versus HIPEC-gemcitabine [24]. A total of 21 patients were enrolled in each arm. No statistically significant differences were observed in postoperative complications (*p* = 0.804), or in >30-day mortality (*p* = 0.999). There was no statistical difference in median OS, median DFS, or distant recurrence between the two groups, though there was a trend toward worse median OS in the HIPEC group at 17.1 months compared to 18. However, locoregional recurrence was significantly lower in the HIPEC group (n = 11 (52%), resection only; n = 2 (10%), resection + HIPEC; *p* = 0.004). These findings corroborate results from other studies which have suggested that locoregional control may be achieved without significant changes in overall survival. Lastly, the role of pancreatic cancer stem cells (PaCSCs) as a possible contributor to recurrence was investigated in the context of HIPEC. PaCSCs are capable of regeneration and differentiation, with the ability for malignant transformation. They have been shown to be resistant to traditional chemotherapy and are associated with worse prognosis. After HIPEC, the number of PaCSCs was found to be significantly reduced (*p* = 0.018), potentially contributing to the lower rates of locoregional recurrence.

Notably, the patients in the Sugarbaker et al. study (HIPEC + NIPEC) were treated with intraperitoneal chemotherapy alone. They did not receive systemic chemotherapy. It is unclear if the patient who only finished three of six planned NIPEC cycles underwent systemic chemotherapy. Similarly, the patients in the Yurttas et al. study received intraperitoneal chemotherapy alone. In the Tentes et al. study, stage III patients received adjuvant systemic chemotherapy (n = 18, 46.2%), while in the phase II/III trial, patients from both arms were considered for individualized adjuvant treatment, with 12 of the 21 patients (57%) in the experimental arm receiving systemic chemotherapy after HIPEC. Variations in intraperitoneal chemotherapy protocol, i.e., 1000 mg/m^2^ gemcitabine for 60 min versus 120 mg/m^2^ gemcitabine for 30 min, also contribute to heterogeneity in study design.

### 2.2. Cytoreductive Surgery and Heated Intraperitoneal Chemotherapy (CRS/HIPEC): Adjuvant Therapy in the Setting of Resected Peritoneal Metastasis

A few studies have explored the role of CRS and HIPEC in patients with PDAC and peritoneal-only metastasis (Table 2).

In the first study, the mean PCI was 12.8 (range 3–25), with ascites present in four cases [26]. Only patients with pancreatic tail tumors were included. Six patients underwent CRS + HIPEC, for a total of eight attempted HIPEC cases. Complete cytoreduction (CC-0) was achieved in five of six patients that underwent CRS/HIPEC, and near-complete cytoreduction (CC-1) in one patient. Distal pancreatectomy was performed in four cases, with two patients having undergone prior pancreatectomy, and one requiring additional resection to reach CC-0. After cytoreduction, half of patients received gemcitabine HIPEC (1000 mg/m^2^) and half were perfused with cisplatin (50 mg/m^2^) + mitomycin C (15 mg/m^2^). Complications were discussed narratively, with one mortality in the postoperative period secondary to liver failure, and no returns to the operating room. One case of pancreatic fistula was observed. Recurrence occurred in five patients, with a mean time of 11.8 months. Three patients developed regional failure, while two developed distant relapse. The final patient had no evidence of disease at the time of publication.

The Mayo Clinic has carried out several studies on CRS/HIPEC for localized PDAC. These investigations evaluate a highly select group of PDAC patients with low-volume PM and/or positive cytology, specifically with a PCI < 7, and at least 6 months of systemic chemotherapy with response before laparoscopic HIPEC [27,28]. Patients proceeded to open CRS/HIPEC only if they were amenable to complete cytoreduction. The HIPEC protocol consisted of perfusion with cisplatin (200 mg/m^2^) + mitomycin C (30 mg/m^2^) for 60 min (phase I pilot study), which was later changed to cisplatin (100 mg/m^2^) and paclitaxel (175 mg/m^2^) for 90 min.

In the initial pilot study of 18 patients, definitive management of the primary tumor depended on resectability [27]. Those who were unable to be resected without pancreatoduodenectomy or vascular reconstruction underwent IRE/IORT (irreversible electroporation/intraoperative radiation therapy) and CRS/HIPEC (n = 11/18). Resectable patients with body/tail tumors underwent distal pancreatectomy and CRS/HIPEC (n = 7/18). All patients in the preliminary cohort achieved CC-0 resection. Two patients (11%) developed CTCAE grade III or higher hematologic toxicity. One postoperative mortality occurred on day 8 due to massive pulmonary embolus, with history of prior hypercoagulability. A total of 71% of patients undergoing pancreatectomy developed grade B pancreatic fistulas, which were managed with drains. Short-term oncologic outcomes include a 1-year OS of 67%. Median follow up was 16 months, with a median PFS of 20 months and median OS of 26 months.

A retrospective study assessing oncologic outcomes followed, which included 61 patients—23 who had undergone CRS/HIPEC compared to 38 who had systemic chemotherapy alone [28]. Median PCI was 2 (IQR 0–4). Complete cytoreduction (CC-0) was achieved in 21patients (91%). There was a statistically significant difference in median overall survival (from PM diagnosis) between the two groups (19 months for systemic alone and 41 months for CRS/HIPEC (*p* = 0.002)). Additionally, 1, 2, and 3-year overall survival was 81%, 31%, and 8%, respectively, for systemic therapy alone, compared to 91%, 66%, and 59% for CRS/HIPEC. From the time of CRS/HIPEC, median OS was 26 months, with 1, 2, and 3-year overall survival of 76%, 57%, and 39%, respectively. Median PFS was 13 months, with 17 months PFS in CC-0 patients (n = 21), and 5 months in CC-1 patients (n = 2) (*p* = 0.001). No patient developed local-only recurrence. Such results suggest a select group of patients with PDAC and isolated PMs may benefit from locoregional treatment with CRS/HIPEC.

These results are from a pilot study preceding a larger phase II clinical trial currently enrolling to evaluate the role of CRS/HIPEC in this selected subset of patients (NCT04858009).

#### Ongoing Studies

The phase II clinical trial from the Mayo Clinic specifically focuses on PDAC patients with low-volume, limited PM or positive cytology (NCT04858009). Patients with PCI of 7 or below and who are deemed to have high success of complete cytoreduction will undergo HIPEC with nab-paclitaxel in a single-arm study. The primary objectives are to evaluate overall survival and disease-free survival, with morbidities from the procedure as a secondary outcome measure.

### 2.3. Pressurized Intraperitoneal Aerosolized Chemotherapy (PIPAC)

A promising alternative to HIPEC is pressurized intraperitoneal aerosolized chemotherapy (PIPAC), which uses a nebulizer to convert liquid chemotherapy into an aerosol [29]. The treatment is performed during standard laparoscopy and the pressure gradient generated by the pneumoperitoneum allows for deeper penetration of the tumor and surrounding tissues [29]. Aerosolization allows for more uniform distribution of intraperitoneal chemotherapy and minimally invasive access makes it simple to repeat treatments every 4–8 weeks [29]. Repeat laparoscopy allows for gross evaluation of tumor response and repeated sampling of PM and normal peritoneum. Importantly, unlike HIPEC, PIPAC is typically not paired with cytoreductive surgery.

The safety and feasibility of PIPAC have been documented in phase I studies [30]. Efficacy in pancreatic cancer has also been suggested, with initial evidence of carcinoma regression in 50–80% of patients [15]. The published literature includes six studies that reported on patients with pancreatic cancer and PM treated with PIPAC. This includes approximately 77 patients (Table 3). Most patients were selected for PIPAC primarily after failing systemic chemotherapy, with extraperitoneal metastases allowed in some studies. PIPAC with cisplatin (7.5 mg/m^2^) and doxorubicin (1.5 mg/m^2^) (PIPAC-CD), or oxaliplatin (92 mg/m^2^) (PIPAC-Ox) for 30 min were the most common regimens. Treatment response was reported via histological regression, commonly with change in the peritoneal regression score (PRGS) introduced by Solass et al. [31,32].

In an early study, Graversen et al. reported on five patients with PMs from PDAC who underwent 16 PIPAC procedures with cisplatin and doxorubicin [33]. Only patients without bowel obstruction and no extraperitoneal metastases were included. All patients had first-line palliative systemic chemotherapy (gemcitabine, S-1). One patient underwent bidirectional therapy, i.e., with concurrent systemic chemotherapy and PIPAC. One patient had a prior distal pancreatectomy, and one a prior total pancreatectomy; three patients had in situ tumors when diagnosed with PM. PCI was recorded at first PIPAC but was not reported. The median OS was 14 months, and four out of five patients achieved histological regression. This was calculated from first PIPAC to last PIPAC regardless of the number of PIPAC procedures. Postoperative complications included transient nausea and vomiting.

Khosrawipour et al. performed a prospective study involving 20 patients with PDAC and PM, with 10 patients eligible for analysis of histologic regression (having undergone two or more PIPAC procedures) [34]. All had prior systemic chemotherapy with variable regimens and subsequently underwent PIPAC-CD. Distinct from the two prior studies, 4 of the 20 patients in this cohort had extraperitoneal metastases—three lung metastases, and one liver. Five of the twenty patients received bidirectional therapy. PCI was 26.6 at first PIPAC. Of the 10 patients that had multiple PIPAC treatments and had a response that was able to be assessed, two patients were found to have complete tumor regression (20%), five had high-grade tumor regression (50%), and the remaining three had no histologic response to PIPAC (30%). Overall, 7/20 (35%) patients had objective histological tumor regression. Median OS was 9 months from first PIPAC. No CTCAE grade III or IV complications occurred. One patient expired on postoperative day 11 from the first PIPAC due to a small bowel obstruction. The procedure-related mortality rate was therefore 2.4%. Unique to PIPAC, failure to gain safe access to the abdominal cavity can occur from severe adhesions and was reported in three patients (7.3%).

In a study including patients with PDAC and cholangiocarcinoma, six patients with PDAC and PM similarly underwent PIPAC with cisplatin and doxorubicin [35]. Patients with peritoneal-only metastasis underwent PIPAC-CD. PCI at first PIPAC for PDAC patients was 13.6. One patient did not have tumor-specific surgery or prior chemotherapy; all others had palliative systemic chemotherapy. Median OS after the first PIPAC procedure was 12.7 months. Complete histological response was induced in two of the six PDAC patients by the second PIPAC procedure, and both were scheduled for further infusions at the time of publication. There were no CTCAE grade III or IV complications. Secondary non-access did not occur, and the authors advocate using Veress needle or mini-laparotomy for safe entry.

In the largest cohort of PDAC patients undergoing bidirectional PIPAC, Di Giorgio et al. evaluated 20 patients with PDAC or cholangiocarcinoma and PMs [36]. Fourteen patients had PDAC. In this subset of patients, six had prior surgery and all had prior systemic chemotherapy. Eight patients had pancreas head lesions and six had tail lesions. Median PCI at first PIPAC was 20. Eleven patients were treated with bidirectional therapy. PIPAC infusion agents varied and were selected based on response to prior systemic chemotherapy. Five patients received PIPAC-CD and nine had PIPAC-Ox. Histologic regression occurred in 50% of patients; specifically, the PIPAC-Ox response rate was 42% and PIPAC-CD was 62%. Median OS was 9.7 months from first PIPAC and 16.2 months from PM diagnosis. No CTCAE grade III or IV complications occurred, and the abdomen was accessed in all cases. Small bowel perforation (<1 cm) occurred upon laparoscopic entry in one patient, which was repaired and did not prevent PIPAC administration.

A follow-up study to results from Graversen et al. used next-generation sequencing (NGS) on representative peritoneal biopsies and peritoneal fluid taken before and after PIPAC to detect changes in mutational profile in response to treatment [37]. This was combined with PRGS to assess histological response. Sixteen patients were included in the study, with five included in the prior publication. Patients had at least one line of palliative systemic chemotherapy, with six patients undergoing bidirectional therapy. A maximum of one extraperitoneal metastasis was allowed. Thirteen patients had undergone at least two PIPAC procedures and were evaluated for histological regression. A total of 61.5% (n = 8) displayed a reduction in mean PRGS and 30.77% (n = 4) had stable disease. The median OS from PIPAC 1 was 9.9 months. On NGS, 15 of 16 patients had a mutation, with KRAS present in 14/16 patients. KRAS mutations in PMs occurred at similar frequencies as in the primary pancreatic site, which may prove useful in predicting mutations if only post-PIPAC biopsies or cytology are available.

Results from the PIPAC-OPC2 study, a phase II trial evaluating use of PIPAC in GI and gynecological malignancies, included a PDAC subgroup of 21 patients [38]. Due to the heterogeneity of the patient population, specific characteristics of the PDAC patients were not reported, but it appears patients underwent PIPAC-CD and could be included in the study with a maximum of one extraperitoneal metastasis. Median OS from PIPAC 1 in the entire group of 110 patients was 10 months and 8.2 months in the PDAC subgroup. From diagnosis of PM, median OS was 19.3 months for all patients and 15.6 months for those with PDAC. While individual data were not available for PDAC patients, the study found that a complete or major histological response was achieved in 38 (61%) of the patients who underwent three PIPAC procedures and was an independent prognostic factor.

A systematic review evaluated the use of PIPAC with bidirectional therapy across multiple cancers [39]. Specifically for PDAC, a total of 7 out of 26 patients (27%) received bidirectional treatment. At the time of publication, the authors found bidirectional treatment to be safe and feasible, but without comparisons to only PIPAC or only systemic treatment. Due to variability in studies and malignancy types, a meta-analysis was unable to be performed.

#### Ongoing Studies

Currently enrolling clinical trials investigating the role of PIPAC in the management of PM in PDAC include three international studies, two of which are phase II single-arm studies, and one phase I dose-escalation study (Table 4). In a study from the University of Hong Kong, patients with PMs from unresectable colorectal cancer, gastric cancer, and PDAC will be treated with bidirectional therapy, i.e., PIPAC (doxorubicin and cisplatin for PDAC) and systemic chemotherapy (NCT06367270). Patients will also be evaluated for downstaging towards resectability and may undergo CRS/HIPEC. Trial objectives include OS, DFS, morbidity, therapeutic efficacy, and complications.

Another, Italian trial will also examine bidirectional therapy with PIPAC, but with nab-paclitaxel as the PIPAC agent (NCT05371223). Systemic therapy will be with gemcitabine and nab-paclitaxel, and trial objectives will include disease control rate using RECIST. This trial will only include patients with PM from PDAC and not other gastrointestinal malignancies.

Lastly, a dose-escalation study is underway in Belgium, where PIPAC with nanoliposomal irinotecan (Nal-Iri) will be used to treat patients with PM from multiple GI malignancies including the pancreas (NCT05277766). The main objective is to find the maximally tolerated dose of drug for a phase II trial, but the trial will also evaluate other efficacy endpoints.

It should be noted that no PIPAC nebulizers have received FDA approval for chemotherapy delivery in the US, though a first-in-the-nation multi-site phase I clinical trial is underway (NCT04329494) [40,41]. The study includes patients with gastrointestinal malignancies, i.e., with gastric, colorectal, and appendiceal primary tumors, but does not include PDAC.

### 2.4. Normothermic Intraperitoneal Chemotherapy (NIPEC)

Normothermic intraperitoneal chemotherapy has been discussed above in the context of HIPEC with subsequent indwelling peritoneal port placement for NIPEC. For PM from PDAC, several studies have reported findings from NIPEC.

NIPEC combined with systemic chemotherapy has been shown to have antitumor activity in those with refractory disease [12]. Intraperitoneal paclitaxel (PTX) combined with S-1, an oral fluoropyrimidine derivative containing tegafur, gimestat, and otastat potassium, is a common regimen [42]. In an early study, patients with refractory pancreatic cancer and malignant ascites underwent both intravenous (50 mg/m^2^) and intraperitoneal PTX (20 mg/m^2^) on days 1 and 8, and S-1 (80 mg/m^2^) on days 1–14 every 3 weeks [12]. Eight of 26 patients evaluable for efficacy achieved a negative change in cytological status, with a disease control rate of 69%. Median OS was 4.8 months and PFS was 2.8 months. Catheter-related infections in two patients (6%) were the only intraperitoneal chemotherapy-related complications. A multicenter phase II study was then conducted with the same regimen, with 7 patients with pancreatic head tumors and 26 in the tail [43]. No distant metastases were allowed. The median survival was 16.3 months, with a 1-year survival of 62%. Severe treatment-related adverse events included anaphylaxis, severe mucositis, diarrhea, and one death from superior mesenteric arterial thrombosis. One patient developed a catheter-related infection, and two more had dislodgement of their device. A multicenter, randomized phase III trial comparing this regimen (IP S1-PTX) to current standard-of-care therapy for metastatic PDAC (gemcitabine plus nab-paclitaxel (GnP)) is currently underway [44]. Patients with presence of distant metastases aside from the ovaries, or microscopic peritoneal dissemination in patients with resectable/borderline resectable PDAC, will be excluded.

Several studies have also evaluated intraperitoneal paclitaxel with intravenous GnP. A phase I trial determined recommended doses for this combination in chemotherapy-naive PDAC patients with PM, with findings of 30 mg/m^2^ of intraperitoneal PTX, 1000 mg/m^2^ of GEM, and 125 mg/m^2^ of nab-PTX, on days 1, 8, and 15 in 4-week cycles [45]. Cytology obtained by peritoneal lavage converted to negative in eight patients (67%), with a PFS of 5.4 months. Port-related complications included obstruction requiring re-implantation (17%), ascites leakage requiring re-suture (8%), and infection requiring re-implantation (8%). Similar results were obtained in a phase I/II study with slightly lower dosing of 20 mg/m^2^ PTX, 800 mg/m^2^ GEM, and 75 mg/m^2^ nab-PTX [46]. Conversion to negative cytology occurred in 18 of 46 patients (39%), with median OS of 14.5 months and 1-year survival of 61%. One CTCAE grade III port complication was observed.

## 3. Discussion

Despite improvements in multidisciplinary cancer care, treatment of PM from PDAC remains a clinical challenge. As early PMs are occult on imaging, the scope of the problem is often underestimated, and patients are excluded from clinical trials. Furthermore, there is a lack of understanding about the distinctiveness of PM biology. Current treatment options are limited to systemic therapy, with limited efficacy and potential for significant toxicity [8]. PMs are more frequently symptomatic than metastases at other sites, which significantly impacts quality of life and further limits tolerability of systemic therapy [4]. The lack of treatment options and loss of hope combines both physical and psychological suffering which contributes to an overall nihilistic approach in this patient population [47]. Novel approaches are therefore greatly needed in which clinicians can take a more proactive approach rather than waiting for an inevitable decline manifested in a gastrointestinal or genitourinary obstruction, ascites, or abdominal pain.

In addition to novel approaches, a reliable method to identify this subset of patients is needed. Positive cytology is known to portend worse survival [48]. According to the National Comprehensive Cancer Network (NCCN), resection is not recommended for patients with localized PDAC, but who have positive cytology: “positive cytology from washings obtained at laparoscopy or laparotomy is equivalent to M1 disease. If resection has been done for such a patient, the patient should be treated for M1 disease”. However, most high-volume centers do not routinely incorporate peritoneal washings in the resectable setting. Presumably, this is explained by the lack of effective treatment options for such patients and the preference to give patients “the benefit of the doubt”. Even in those who do not have positive cytology initially, tumor manipulation during resection induces dissemination of tumor cells into the peritoneal cavity in up to one third of patients [49], with ongoing studies incorporating extensive washing after resection to determine whether lower rates of peritoneal relapse can be achieved [50]. In gastric cancer, staging laparoscopies are routinely recommended and shift initial therapeutic management from upfront surgery to systemic treatment in the presence of metastases [51]. However, in PDAC, NCCN guidelines recommend diagnostic laparoscopy in patients with CA 19–9 level > 150 U/mL, low-volume ascites, tumor in the body of the pancreas, borderline resectable tumor, tumor size >3 cm, and common bile duct lymphadenopathy. Despite these recommendations, diagnostic laparoscopy is still underutilized in clinical practice.

Up to 18% of patients undergoing staging laparoscopy in PDAC have metastases or positive cytology, with an increase to 23% in those who have not had prior neoadjuvant chemotherapy [52]. As the repertoire of systemic and peritoneal-directed chemotherapy expands and improves, identification of this subset of patients may be crucial for incorporation of regional approaches that may result in a more favorable prognosis while preserving quality of life.

Intraperitoneal chemotherapy provides an approach to regional therapy that can augment systemic chemotherapy. While data have been promising in other gastrointestinal malignancies, with evidence of increased overall survival depending on primary tumor type, there has been a lack of enthusiasm in the context of PDAC due to paucity of data and overall poor outcomes [53]. Discrepancies in patient selection and procedural considerations related to not only the administration of intraperitoneal chemotherapy, but also the subsequent use of adjuvant systemic therapy result in an abundance of regimens without a clear consensus.

The incorporation of HIPEC in PDAC falls into three broad categories: (1) primary prevention of PM in those with localized disease at the time of resection, (2) patients with low-burden PMs and positive cytology amenable to CRS, and (3) laparoscopic HIPEC repeated at select intervals with the goal of downstaging peritoneal metastasis or positive cytology. For the former, there is some signal for HIPEC efficacy in reducing locoregional failure, but high rates of distant metastasis ultimately drive overall survival. In patients with resectable low-volume PM, studies evaluating CRS/HIPEC highlight the importance of achieving a complete cytoreduction: 1, 2, and 3-year OS from CRS/HIPEC for CC-0 was 80%, 60%, and 41%, respectively, compared to 0% for all with CC-1 [28]. Similarly, 1, 2, and 3-year PFS from CRS/HIPEC for CC-0 was 66%, 37%, and 37%, respectively, compared to 0% for all with CC-1. These results are promising when compared to chemotherapy alone, with only 8% of patients alive at 3 years after diagnosis of PMs, compared to 59% who were treated with both systemic chemotherapy and HIPEC/CRS. Ultimately, HIPEC appears to be safe and feasible in the setting of pancreatectomy, with a possible signal of improved local control in the setting of a small group of patients. In a select group of initially unresectable patients due to low-volume PM, HIPEC may be able to downstage disease for complete cytoreduction and prolong DFS, but appropriate assessment of impact on long-term survival benefit awaits ongoing and future randomized clinical trials.

PIPAC is a novel approach to peritoneal chemotherapy delivery with a few potential applications in PDAC patients. One area of active investigation includes patients with peritoneal PM or positive cytology from PDAC who are candidates for “bidirectional therapy” with combined standard-of-care systemic therapy and PIPAC at 4–8 week intervals. Another potential application includes utilization in the adjuvant setting, where PIPAC is delivered under the same anesthesia as minimally invasive pancreatectomy. Most early studies exploring PIPAC include a heterogenous group of patients with varying tumor types, extents of peritoneal disease, and systemic regimens, which makes it impossible to draw conclusions about efficacy. Furthermore, the field is yet to define an optimal efficacy endpoint. In most studies, treatment response is assessed in multiple ways, including the peritoneal regression score (PRGS), which has been widely adopted since its introduction by Solass and colleagues. Studies vary on selecting mean versus median PRGS for the biopsies taken per laparoscopic procedure, with some recent evidence suggesting that median PRGS combined with presence of positive or negative cytology may have better prognostication than using mean PRGS or PRGS alone [54]. Measuring changes in PRGS between PIPAC procedures also vary, from the first PIPAC to the last PIPAC or between a set number of procedures. Additional efficacy endpoints include radiographic response by RECIST criteria, biochemical response with tumor markers, circulating tumor DNA, circulating peritoneal DNA, quality of life [55], progression-free and overall survival. Importantly, patients with advanced PM often have undetectable circulating tumor DNA [56,57].

Despite these limitations, there are promising signals of tumor response. Multiple studies show surprising objective response rates of 50–80%, with additional data indicating that complete pathologic response is possible with limited disease burden. As many patients have exhausted routine systemic therapies, PIPAC could be used to augment chemotherapy in a bidirectional approach or act as a unidirectional locoregional treatment for patients who cannot tolerate or are no longer responsive to systemic chemotherapy. Importantly, PIPAC is well tolerated and may allow patients with terminal illness to maximize quality of life while still prolonging disease control. Overall, PIPAC appears to be safe and feasible, with the potential to induce histologic response. However, definitive improvements in overall survival remain elusive, with a need for controlled, prospective trials to draw further conclusions with standardized chemotherapy regimens and patient populations.

Finally, it is important to note that HIPEC, PIPAC, and NIPEC are mechanisms of drug delivery, and efficacy depends on selection of the proper therapeutic agent. As systemic therapy continues to improve, novel agents should also be studied in the peritoneum. Access to longitudinal sampling using PM technologies may offer an opportunity to study disease biology and tumor response to further advancements in translational research.

## 4. Conclusions

Peritoneal metastases in patients with PDAC are understudied. Their presence drives survival and quality of life. Further study is needed to understand their unique biology and determine opportunities for novel therapeutics that target peritoneal metastasis. The utility of cytoreductive surgery and intraperitoneal chemotherapy is controversial and unproven in patients with PDAC, but there may be a role for these regional approaches in the palliative and curative-intent/adjuvant setting. Standardization of treatments and evaluations of response are needed to define the optimal endpoints for efficacy.

## Figures and Tables

**Figure 1 curroncol-32-00103-f001:**
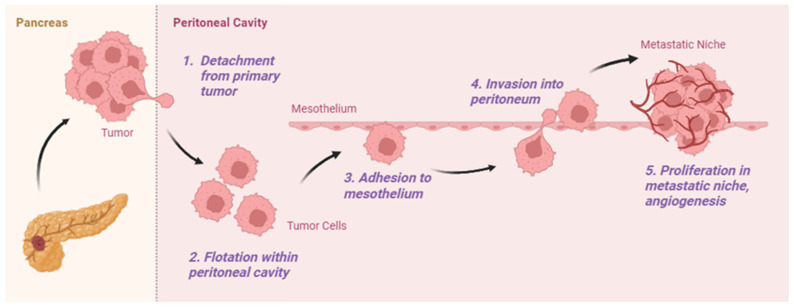
Metastatic cascade of peritoneal metastasis which includes (1) detachment from primary tumor, (2) flotation within peritoneal cavity, (3) adhesion to mesothelium, (4) invasion into peritoneum, and (5) proliferation in metastatic niche with angiogenesis [10].

**Figure 2 curroncol-32-00103-f002:**
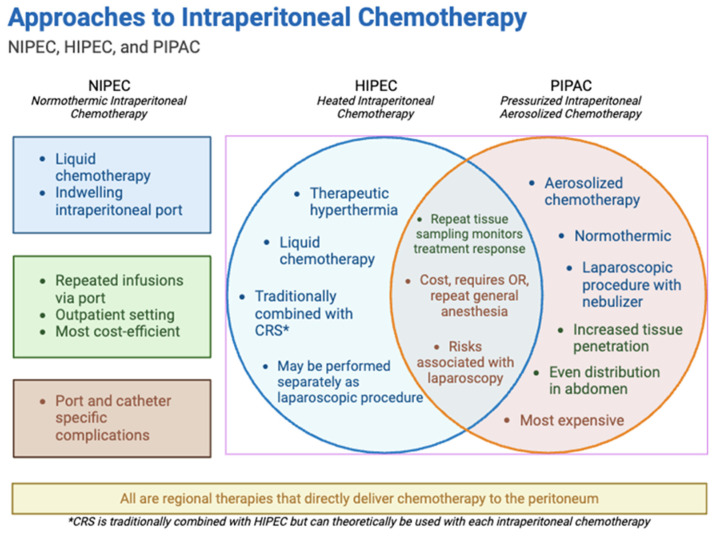
Approaches to NIPEC, HIPEC, and PIPAC.

**Figure 3 curroncol-32-00103-f003:**
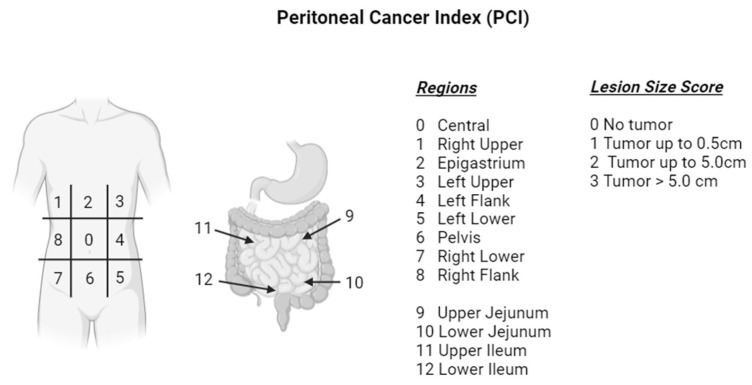
The peritoneal cancer index (PCI), first introduced by Sugarbaker in 1995, quantifies the spread of peritoneal carcinomatosis by estimating tumor burden in 12 sections. The abdomen is divided into 9 areas, and the small bowel into 4 additional sectors. A lesion score (0–3) is assigned to each section based on tumor size for a sum of 0–36.

**Table 1 curroncol-32-00103-t001:** Heated Intraperitoneal Chemotherapy (HIPEC): Adjuvant Therapy in the Setting of Resected Localized Disease.

Author	n	Treatment	HIPEC Agent	Primary Endpoints	Survival	Recurrence
Sugarbaker, 2020 [21]	12	Resection + HIPEC; adjuvant NIPEC for 6 cycles	Gemcitabine (1000 mg/m^2^) for 60 min	Safety Pharmacokinetics	mOS: 29 months	None at time of publication
Yurttas, 2021 [22]	16	Resection + HIPEC; no adjuvant systemic chemotherapy	Gemcitabine (1000 mg/m^2^) for 60 min	30-day mortality	mOS: 17 months mPFS: 11 months	Not reported
Tentes, 2021 [23]	39	Resection + HIPEC; adjuvant systemic chemotherapy for stage III patients	Gemcitabine (1000 mg/m^2^) for 60 min	Morbidity, mortality Survival Recurrence	mOS: 16.1 months 1-year survival: 62.5%	Recurrence (n = 23, 59%) Locoregional failure (n = 4, 10.3%) Distant disease (n = 19, 48.7%)
Padilla-Valverde, 2024 [24]	21	Arm I: Resection (n = 21) ± adjuvant systemic chemotherapy (n = 16) Arm II: Resection + HIPEC (n = 21) ± adjuvant systemic chemotherapy (n = 12)	Gemcitabine (120 mg/m^2^) for 30 min	Recurrence OS DFS Pancreatic cancer stem cells	mOS: 18 months (resection only) 17.1 months (resection + HIPEC) *p* = 0.899 mDFS: 10 months (resection only) 14 months (resection + HIPEC) *p* = 0.888	Locoregional recurrence: n = 11, 52% (resection only) n = 2, 10% (resection + HIPEC) *p* = 0.004 Distant recurrence: n = 8, 38% (resection only) n = 9, 43% (resection + HIPEC) *p* = 0.757

n: number of patients who underwent HIPEC; mOS: median overall survival; mPFS: median progression-free survival; mDFS: median disease-free survival.

**Table 2 curroncol-32-00103-t002:** Heated Intraperitoneal Chemotherapy (HIPEC): Adjuvant Therapy in the Setting of Resected Peritoneal Metastasis.

Author	n	Treatment	HIPEC Agent	Primary Endpoints	Survival	Recurrence
Tentes, 2018 [26]	6	CRS + HIPEC + adjuvant systemic chemotherapy	Gemcitabine (1000 mg/m^2^) for 60 min OR Cisplatin (50 mg/m^2^) + mitomycin C (15 mg/m^2^)	Descriptive institutional experience	1-year survival 67% 2-year survival 16%	Recurrence (n = 5, 83%) Locoregional (n = 3, 50%) Distant disease (n = 2, 33%)
Grotz, 2023 [27]	18	CRS + HIPEC Management of primary tumor: IORT (n = 1) IRE (n = 10) Formal resection (n = 7)	Cisplatin (200 mg/m^2^) + mitomycin C (30 mg/m^2^) for 60 min	1-year survival PFS OS Surgical outcomes	1-year survival: 67% mOS: 26 months mPFS: 20 months	Total (n = 8, 44%) Peritoneum (n = 5, 45%) Liver (n = 2, 11%) Distant site (n = 1, 6%)
Gudmundsdottir, 2023 [28]	23	CRS + adjuvant systemic chemotherapy (n = 38) CRS + HIPEC (n = 23) No adjuvant systemic chemotherapy after CC-0 resection (n = 21)	Cisplatin (200 mg/m^2^) + mitomycin C (30 mg/m^2^) for 60 min OR Cisplatin (100 mg/m^2^) and paclitaxel (175 mg/m^2^) for 90 min	1, 2, 3-year PFS from CRS/HIPEC 1, 2, 3-year OS from PM diagnosis	1, 2, 3-year PFS (60%, 33%, 33%) 1, 2, 3-year OS (76%, 57%, 39%) mOS from CRS/HIPEC: 26 months mOS from PM diagnosis: 31 months (CRS/HIPEC) 19 months (systemic alone) *p* = 0.002	Total (n = 12; 52%) Peritoneum (n = 8; 35%) Liver (n = 2; 9%) Lung (n = 2; 9%)

n: number of patients who underwent HIPEC; IORT: intraoperative radiation therapy; IRE: irreversible electroporation; mOS: median overall survival; mPFS: median progression-free survival.

**Table 3 curroncol-32-00103-t003:** Pressurized Intraperitoneal Aerosolized Chemotherapy (PIPAC) in the Setting of Unresectable Disease with Peritoneal Metastasis.

Author	n	E-PM	Prior PSC	PIPAC Agent	Systemic Agent	Histological	Median OS
Graversen, 2017 [33]	5	None	100%	Cisplatin (7.5 mg/m^2^) + Doxorubicin (1.5 mg/m^2^) for 30–35 min	Gemcitabine + S-1 (n = 1, 20%)	PRGS Highest/mean, first to last PIPAC Regression: n = 4 (80%)	PIPAC: 6 months PM diagnosis: 14 months
Khosrawipour, 2017 [34]	20	4 (20%)	100%	Cisplatin (7.5 mg/m^2^) + Doxorubicin (1.5 mg/m^2^) for 30 min	Gnp (n = 2; 10%) Gemcitabine (n = 2; 10%) FOLFIRINOX (n = 1, 5%) Total (n = 5; 25%)	TRG Any regression with ≥2 PIPAC cycles Regression: n = 7 (35%)	PIPAC: 9.2 months
Horvath, 2018 [35]	6	None	5 (83%)	Cisplatin (7.5 mg/m^2^) + Doxorubicin (1.5mg/m^2^) for 30 min	None	PRGS Highest/mean, Any regression with ≥2 PIPAC cycles Regression (complete): n = 2 (33%)	PIPAC: 12.7 months
Di Giorgio, 2020 [36]	14	None	100%	Cisplatin (7.5 mg/m^2^) + Doxorubicin (1.5mg/m^2^) for 30 min OR	Mixed Total (n = 11; 85%)	PRGS Highest/mean, Any regression with ≥2 PIPAC cycles	PIPAC: 9.7 months
				Oxaliplatin (92 mg/m^2^) if good response to prior PSC, or severe side effects to CD		Regression: n = 7 (50%)	
Nielsen, 2021 [37]	16 **	Max 1 allowed	100%	Cisplatin (7.5 mg/m^2^) + Doxorubicin (1.5 mg/m^2^) for 30 min	n = 4, 25%	PRGS Highest/mean, Any regression with ≥2 PIPAC cycles Regression: n = 8 (50%)	PIPAC: 9.9 months
Graversen, 2023 [38]	21	Max 1 allowed	***	Cisplatin (7.5 mg/m^2^) + Doxorubicin (1.5 mg/m^2^) for 30–35 min	*	PRGS Highest/mean, PIPAC 1–3 ***	PIPAC: 8.2 months

n: number of patients who underwent PIPAC; E-PM, extraperitoneal metastasis; Prior PSC: prior palliative systemic chemotherapy. * = data reported in aggregate; ** = 5 patients included in prior study [32].

**Table 4 curroncol-32-00103-t004:** Ongoing Studies in HIPEC and PIPAC.

Author/Instit.	Design	Population	Treatment	Status	Objectives
Thiels, Mayo Clinic, Rochester (NCT04858009)	Phase II, single arm	PM from PDAC—limited low volume or positive cytology	HIPEC, nab-paclitaxel and cisplatin	Enrolling (1/25/2024); estimated completion (5/2026) 40 patients	(1) OS, DFS (2) Morbidity
Wong, U of Hong Kong (NCT06367270)	Phase II, single arm	PM from unresectable CRC, gastric cancer, and PDAC	Bidirectional PIPAC doxorubicin + cisplatin (gastric, pancreatic), oxaliplatin (CRC); May downstage and undergo CRS/HIPEC if becomes resectable	Enrolling (09/01/2023); estimated completion (08/31/2027) 60 patients	Therapeutic efficacy, complications (1) Clinical benefit rate measured by RECIST, PCI, and histopathology (2) Adverse events, PFS, OS
Di Giorgio, Fondazione Policlinico Universitario Agostino Gemelli IRCCS, Rome (NCT05371223)	Phase II, single arm	PM from PDAC	Bidirectional (systemic GnP), PIPAC nab-paclitaxel	Enrolling (03/01/2022); estimated completion (07/30/2025) 38 patients	(1) Disease control rate using RECIST
Ceelen, U of Ghent, Belgium (NCT05277766)	Phase I, single arm	PM from multiple GI cancers, includes PDAC	PIPAC, dose-escalation study with nanoliposomal irinotecan (Nal-IrI)	Enrolling (11/21/2022); estimated completion (01/01/2027) 45 patients	(1) Maximally tolerated dose (2) Recommended phase II dose, morbidity, pharmacokinetic parameters

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
