# Peer review of "Management of Peritoneal Metastasis in Patients with Pancreatic Ductal Adenocarcinoma"

_curroncol, 2025, doi:10.3390/curroncol32020103_

Round 1

Reviewer 1 Report

Comments and Suggestions for Authors

This is a comprehensive and well-written narrative review of current data for management of peritoneal carsinomatosis/metastasis in PDAC. The authors of this study have balanced the description of characteristics and distinctive features of pancreatic cancer with peritoneal metastasis, with a thorough review of available data on efforts to treat this entity. The figures and tables are good, and adequately chosen for the purpose of this review. The data is presented in a structured manner; not so much summarized since the studies are heterogeneous and not well suited for meta-analysis, but presented very clearly. The discussion is also good, presenting the relevant dilemmas in this fiels. And the conclusion is sober and consistent with the knowledge that has been derived from existing data.

I do not have any specific suggestions for improvement, except that the authors might explain what is meant by the following sentence (line 208): "The Mayo Clinic has carried out a prospective study, retrospective study, and is currently evaluating a prospective phase 2 trial using CRS/HIPEC for localized PDAC." This particular sentence needs to be more precise.

Otherwise, this manuscript provides an important insight for the reader of the current status of management of peritoneal metastasis in patients with pancreatic duct adenocarcinoma (which is also the title of the manuscript, and thus, the title is also precise).

The abstract is well-written. And previous work is also adequately referenced.

Congratulations to the authors.

Author Response

Comment 1: I do not have any specific suggestions for improvement, except that the authors might explain what is meant by the following sentence (line 208): "The Mayo Clinic has carried out a prospective study, retrospective study, and is currently evaluating a prospective phase 2 trial using CRS/HIPEC for localized PDAC." This particular sentence needs to be more precise.

Response 1: Thank you for the comments. We agree this sentence can be clarified and have made changes in the manuscript (line 230). 

Reviewer 2 Report

Comments and Suggestions for Authors

This is a well written and structured review, highlighting the main problem of PM in PDAC: heterogeneity of the additional treatment of the included patients and low numbers.

Nevertheless, it helps the reader to be updated on this clinically highly relevant topic

Minor:  In the conclusion you may add a sentence about the need for standardizing diagnostic tests (e.g. cytology) in oder to be able to perform in the future studies with less heterogeneity within the included patients.

Author Response

Comment 1: "Minor:  In the conclusion you may add a sentence about the need for standardizing diagnostic tests (e.g. cytology) in oder to be able to perform in the future studies with less heterogeneity within the included patients." 

Response 1: Thank you for the comments. This suggestion has been included in the conclusion (lines 540-541).  

Reviewer 3 Report

Comments and Suggestions for Authors

Summary: Peritoneal metastasis (PM) is a frequent complication in pancreatic ductal adenocarcinoma (PDAC), with up to 50% of patients developing PM after surgery, leading to dismal prognoses. PM is often underdiagnosed due to subtle imaging findings and remains a significant challenge in PDAC management. Additionally, patients with PM are frequently excluded from clinical trials, further limiting therapeutic advancements. Intraperitoneal therapies, such as heated intraperitoneal chemotherapy (HIPEC) combined with cytoreductive surgery (CRS), have demonstrated efficacy in other cancers like colorectal and ovarian cancer but are underutilized in PDAC. Emerging data suggest that select PDAC patients with PM may benefit from these therapies, particularly if predictive biomarkers can be identified to guide treatment decisions.

Strengths: This review highlights a neglected aspect of PDAC treatment by focusing on intraperitoneal therapies for PM, addressing the potential benefits of HIPEC and normothermic intraperitoneal chemotherapy (NIPEC).

Comments:

  1. Discuss the role of EUS and PET-CT in the diagnosis of PM in the introduction.
    • Incorporate a detailed discussion on how endoscopic ultrasound (EUS) and PET-CT scans contribute to detecting subtle signs of peritoneal spread. EUS can identify small-volume PM that may be missed by conventional imaging, while PET-CT offers metabolic insights that can indicate peritoneal disease presence.
  2. How were the adverse events graded for gemcitabine HIPEC and NIPEC therapy?
    • Clarify the grading criteria used for adverse events in studies involving gemcitabine-based HIPEC and NIPEC therapies. Specify whether the Common Terminology Criteria for Adverse Events (CTCAE) was employed and provide examples of adverse events observed.
  3. Please define the adverse events associated with intraperitoneal treatment in detail in the discussion section.
    • Expand the discussion on the types of adverse events linked to HIPEC and NIPEC. These may include abdominal pain, nausea, ileus, peritonitis, and more severe complications like anastomotic leaks or sepsis. Highlight the frequency and management strategies of these complications.
  4. In the study by Tentes et al., were there any patients with PM in the ones with locoregional spread?
    • Provide a comprehensive analysis of the patient cohort in the Tentes et al. study, specifying whether any patients had documented peritoneal metastases among those with locoregional spread. This will help assess the applicability of CRS and HIPEC in this patient population.
  5. Elaborate on the role of pancreatic cancer stem cells (PaCSCs) in the study by Padilla-Valverde et al., particularly in relation to locoregional spread and control by HIPEC.
    • Discuss how PaCSCs contribute to PM and the rationale behind targeting them with HIPEC. Include the mechanisms through which HIPEC may impact the survival and proliferation of these stem cells and reduce locoregional spread.
  6. Discuss the various cut-offs used for peritoneal cancer index (PCI) to determine eligibility for cytoreductive surgery.
    • Explore the different PCI thresholds used in clinical practice and research to identify suitable candidates for CRS. Discuss the implications of high versus low PCI scores on patient outcomes and how they influence treatment decisions.
  7. Provide an algorithm in a figure to explain the various available treatments for patients with PM.
    • Create a visual algorithm that outlines the diagnostic and treatment pathways for patients with PM, including options like systemic chemotherapy, HIPEC, NIPEC, and CRS. The figure should include decision points based on PCI scores, performance status, and biomarker profiles.

Author Response

Comment 1: Discuss the role of EUS and PET-CT in the diagnosis of PM in the introduction. Incorporate a detailed discussion on how endoscopic ultrasound (EUS) and PET-CT scans contribute to detecting subtle signs of peritoneal spread. EUS can identify small-volume PM that may be missed by conventional imaging, while PET-CT offers metabolic insights that can indicate peritoneal disease presence. 

Response 1: Thank you for the suggestion. We agree that it is crucial to be able to reliably detect and quantify PMs for accurate staging and recognition of treatment options that may benefit those with low-volume PM. At this time neither EUS nor PET-CT are routinely used for diagnosis of PM in PDAC due to prevalence of CT for staging. There is no solid evidence that EUS or PET-CT are consistently superior to detecting radiographically “occult” PMs than CT. In fact, there may be some evidence that CT is superior for detection of these lesions, as there is less FDG uptake in smaller lesions in PET-CT, and EUS has been shown to be superior for detecting locoregional invasion, but inferior for distance metastases. The lack of a clear advantage is why we instead advocate for the use of diagnostic laparoscopy in our paper. Diagnostic laparoscopy has been shown to be unequivocally superior in detecting sub-centimeter deposits via direct visualization, while offering the unique opportunity for tissue sampling and collecting baseline characteristics prior to any treatment.  

Comment 2: How were the adverse events graded for gemcitabine HIPEC and NIPEC therapy? Clarify the grading criteria used for adverse events in studies involving gemcitabine-based HIPEC and NIPEC therapies. Specify whether the Common Terminology Criteria for Adverse Events (CTCAE) was employed and provide examples of adverse events observed. 

 Response 2: In regard to gemcitabine specific intraperitoneal therapy, studies varied on reporting complications with CTCAE, the Clavien-Dindo score, or only narratively. We have added these details to their respective paragraphs. The NIPEC studies all used intraperitoneal paclitaxel and not gemcitabine-based therapy, which was administered intravenously.    

Comment 3: Please define the adverse events associated with intraperitoneal treatment in detail in the discussion section. Expand the discussion on the types of adverse events linked to HIPEC and NIPEC. These may include abdominal pain, nausea, ileus, peritonitis, and more severe complications like anastomotic leaks or sepsis. Highlight the frequency and management strategies of these complications. 

Response 3: Thank you for this suggestion. Understanding adverse events associated with intraperitoneal therapy is crucial to evaluating their safety and feasibility. Due to heterogeneity in methodology between studies, we have added general adverse events to each respective study paragraph to the best of our ability.   

Comment 4: In the study by Tentes et al., were there any patients with PM in the ones with locoregional spread? Provide a comprehensive analysis of the patient cohort in the Tentes et al. study, specifying whether any patients had documented peritoneal metastases among those with locoregional spread. This will help assess the applicability of CRS and HIPEC in this patient population. 

Response 4: Thank you for this suggestion. We agree that this is a key distinction that was not previously clear. The Tentes 2021 study reported the cases of locoregional recurrences as T3N1, indicating no overlap with PM development, and we have added this clarification to the manuscript (lines 176-177).  

Comment 5: Elaborate on the role of pancreatic cancer stem cells (PaCSCs) in the study by Padilla-Valverde et al., particularly in relation to locoregional spread and control by HIPEC. 

Discuss how PaCSCs contribute to PM and the rationale behind targeting them with HIPEC. Include the mechanisms through which HIPEC may impact the survival and proliferation of these stem cells and reduce locoregional spread. 

Response 5: Thank you for this suggestion. We agree that more background information on the significance of PaCSCs is important and have expanded on their role in PMs as described by the authors of the study (lines 192-194).  

Comment 6: Discuss the various cut-offs used for peritoneal cancer index (PCI) to determine eligibility for cytoreductive surgery. 

Explore the different PCI thresholds used in clinical practice and research to identify suitable candidates for CRS. Discuss the implications of high versus low PCI scores on patient outcomes and how they influence treatment decisions. 

Response 6: Thank you for this suggestion. We agree that PCI has the potential to aid in clinical decision making and offer insight into patient outcomes. Unfortunately, these numbers have not yet been determined for PMs from PDAC. While PCI has been explored in other gastrointestinal malignancies, it would be premature to extrapolate prognostic factors for PDAC due to differences in underlying tumor biology (i.e. in colorectal cancer a PCI>17 correlates with worse overall survival, but this number is PCI>12 in gastric cancer). The heterogeneity of the few study designs in the current PDAC literature also makes it difficult to recommend distinct PCI cut-offs for when to use CRS. As future clinical trials enroll larger cohorts of patients, we agree that PCI should certainly be an important factor in trial design and data collection. However, given the current literature, it is important to recognize that the discussed technologies are not standard of care, and our paper only aims to highlight active areas of study. At this time, we do not advocate for the routine use of CRS or intraperitoneal chemotherapies for PMs from PDAC in general practice outside of a clinical trial.  

Comment 7: Provide an algorithm in a figure to explain the various available treatments for patients with PM. 

Create a visual algorithm that outlines the diagnostic and treatment pathways for patients with PM, including options like systemic chemotherapy, HIPEC, NIPEC, and CRS. The figure should include decision points based on PCI scores, performance status, and biomarker profiles. 

Response 7: Thank you for this suggestion. Unfortunately, there is not enough research at this time to make evidence-based suggestions for which modality to select based on performance status, PCI, or biomarkers in the context of PDAC derived PMs. These are excellent points and should ideally be pursued in future studies as part of thoughtful clinical trial designs. We agree that a visual tool to differentiate the various intraperitoneal therapies would be helpful. We have added a third figure to compare and contrast intraperitoneal therapies, and a graphical abstract for current treatment options. These aim to clarify the question of which therapies might be available to specific patients, as well as other clinical considerations that may be important upon wider adoption of these technologies.  

Reviewer 4 Report

Comments and Suggestions for Authors

The authors of the manuscript entitled “Management of Peritoneal Metastasis in Patients with Pancreatic Ductal Adenocarcinoma” by Grace Wu et al,  describe in their review the current state of management of patients with peritoneal metastasis from PDAC. In addition, the authors review the available data exploring peritoneal-directed therapy with cytoreductive surgery and/or intraperitoneal chemotherapy. Overall, the manuscript is well researched and well written and is supported with adequate figures and tables throughout their review. This review also provides a strong one stop go to spot to understand the role of peritoneal metastasis in PDAC and the potential treatments available.

Here are some of my comments and minor suggestions on the manuscript.

a)        The authors can include more statistical inputs between the 4 studies used in section 2.1. Curative-Intent Heated Intraperitoneal Chemothreapy (HIPEC): Adjuvant Therapy in the 115 Setting of Resected Localized Disease. This could include important inputs like hazards ratio and significance of the study based on the sample size used for the study and other key findings from easy study. Furthermore, the authors can add an extra column or two in this regard to enhance the comparison. This is a minor suggestion. The authors could also color their data table to group either by commonalities (HIPEC treatment regimen) or differences or rank them based of high statistical significance.

b)       The authors mentioned the treatment regimen of the HIPEC agent. However, it would also be important to mention how the two drug concentrations of 1000mg/m2 and 120mg/m2 and the treatment time influence the outcome of the results. Interpretation of the impact would be highly beneficial to the readers and worthwhile discussing.

c)        I would suggest the authors to make the same changes with respect to their table 2 and table 3 to the HIPEC and PIPAC agents used and bring out the statistical significance between the projects.

d)       Overall, the authors could also work on a figure 3 where they can show the pros and cons for the three therapeutic agents HIPEC, PIPAC, and NIPEC and their mechanisms of drug delivery and criteria for the choice of treatment.

Author Response

Comment 1: The authors can include more statistical inputs between the 4 studies used in section 2.1. Curative-Intent Heated Intraperitoneal Chemothreapy (HIPEC): Adjuvant Therapy in the 115 Setting of Resected Localized Disease. This could include important inputs like hazards ratio and significance of the study based on the sample size used for the study and other key findings from easy study. Furthermore, the authors can add an extra column or two in this regard to enhance the comparison. This is a minor suggestion. The authors could also color their data table to group either by commonalities (HIPEC treatment regimen) or differences or rank them based of high statistical significance. 

Response 1: Thank you for these suggestions. We agree that statistical significance lends important weight to interpretation of study results. Unfortunately, three of the four studies were descriptive studies without a control arm, and statistical analysis could not be performed due to inability to compare two groups. For the clinical trial, we included reported p values in our table as well as in the text. These include overall survival, locoregional recurrence, and distant recurrence. We also agree that visually differentiating the intraperitoneal therapies would aid in understanding. However, even within each study there is heterogeneity in which regimens were used. The complexity would make it difficult to color each study in a helpful way (e.g. in table 2, there are 4 different regimens for 3 studies).  

Comment 2: The authors mentioned the treatment regimen of the HIPEC agent. However, it would also be important to mention how the two drug concentrations of 1000mg/m2 and 120mg/m2 and the treatment time influence the outcome of the results. Interpretation of the impact would be highly beneficial to the readers and worthwhile discussing. 

Response 2: Thank you for pointing this out. We agree this distinction is important and have added emphasis to this in our summary paragraph discussing chemotherapy regimens for HIPEC (lines 204-207). 

Comment 3: I would suggest the authors to make the same changes with respect to their table 2 and table 3 to the HIPEC and PIPAC agents used and bring out the statistical significance between the projects. 

Response 3: Thank you for this suggestion. Similar to the above response, statistical values were provided when applicable, i.e. for the Gudmundsdottir study, where median overall survival was provided from the time of PM diagnosis. 

Comment 4: Overall, the authors could also work on a figure 3 where they can show the pros and cons for the three therapeutic agents HIPEC, PIPAC, and NIPEC and their mechanisms of drug delivery and criteria for the choice of treatment. 

Response 4: Thank you for this suggestion. We agree that an organized visual differentiating these treatments would be helpful. This has been added to the manuscript in the form of a third figure and a graphical abstract.